# Hand-Wrist Disorders in Chainsaw Operators: A Follow-Up Study in a Group of Italian Loggers

**DOI:** 10.3390/ijerph18147217

**Published:** 2021-07-06

**Authors:** Federica Masci, Giovanna Spatari, Concetto Mario Giorgianni, Elisa Pernigotti, Laura Maria Antonangeli, Vittorio Bordoni, Alberto Magenta Biasina, Luca Pietrogrande, Claudio Colosio

**Affiliations:** 1Department of Health Sciences of the University of Milano and International Centre for Rural Health of the Occupational Health Unit of the Hospitals Santi Paolo e Carlo, 20142 Milano, Italy; lantonangeli@outlook.it (L.M.A.); claudio.colosio@unimi.it (C.C.); 2Department of Biomedical, Dental and Morphological and Functional Imaging, University of Messina, 98125 Messina, Italy; giovanna.spatari@unime.it (G.S.); mariogiorgianni@virgilio.it (C.M.G.); 3Post Graduate School in Orthopedics and Traumatology, University of Milan, 20142 Milano, Italy; elisa.pernigotti@unimi.it; 4Post Graduate School in Occupational Medicine, University of Milan, 20122 Milano, Italy; vitto-rio.bordoni@unimi.it; 5Diagnostic and Interventional Radiology School of Santi Paolo and Carlo ASST of Milan, 20142 Milano, Italy; magentabiasina@gmail.com; 6Department of Health Sciences, University of Milan, 20142 Milano, Italy; luca.pietrogrande@unimi.it

**Keywords:** logging, musculoskeletal disorders, wrist, ultrasound investigation, CTS

## Abstract

Despite the mechanization process implemented in arboriculture, logging tasks are still manually performed by chainsaw operators, which therefore are exposed to the risk of developing hand-wrist musculoskeletal disorders. Our research aimed to: (a) define whether the slight changes observed in 2017 showed an evolution to overt diseases; (b) study some risk determinants for these diseases such as age, working experience, and performing a secondary job. We recruited in a two-year follow-up study, 38 male forestry workers performing logging tasks employed in the Sicilian Forestry Department located in Enna. All the subjects underwent: (1) personal data collection; (2) administration of questionnaire addressed at upper limbs symptoms with a hand chart; (3) physical examination of the upper limbs, including Tinel’s and Phalen’s maneuvers; (4) ultrasound investigation of the hand-wrist area. In the two-year follow-up study we registered an overall increasing in wrist disorders, thus we can assume that forestry workers may be a target population for wrist diseases and deserve a particular attention in workers’ health surveillance programs. Interestingly, the prevalence of wrist-hand disorders resulted to be higher in younger workers.

## 1. Introduction

In the period between 2008 and 2012, the reports of occupational musculoskeletal disorders (MSD) in agriculture delivered to the Italian workers’ compensation authority (INAIL) increased from 41 to 65% of the total. [1] Agricultural and forestry workers are particularly at risk of developing MSD compared to other sectors, due to the characteristics of the job tasks and the environmental conditions in which the activities are performed [2,3]. Studies carried out in this sector are mainly focused on typical agricultural activities, whilst forestry is less investigated, despite the fact that it can be considered one of the most dangerous of all the occupations, and can be included in the list of the so called “3D” jobs—dirty, dangerous, and demanding [4]. Among forestry activities, particular attention should be addressed at logging, a sector in which, despite the mechanization of the whole process, part of the activities are still manually performed by workers. The three main tasks manually performed are felling, delimbing, and bucking (Figure 1). The main biomechanical risk factors to whom chainsaw operators are exposed includes hand-arm vibration, awkward postures, repetitive movements, and extreme environmental conditions [5].

Most of the harvesting routine is performed in awkward posture, due to the manual handling of the chainsaw. These activities, together with the handling of heavy loads for long periods, for example, during wood gathering or transport, can lead to a higher risk of MSDs, especially if the work rhythm is too high or if the workers are not trained to follow good working practices [6,7,8]. Risks mainly affect both tendons and joints and spine, in particular, the lumbar tract [9,10].

The few existing studies addressed at the effects of prolonged use of chainsaw show an increased risk of diseases affecting mainly the peripheral neural system, the peripheral vascular system, or the musculoskeletal system, in the so called “hand-arm vibration syndrome” (HAVS) [11,12]. Moreover, working with a chainsaw seems associated with carpal tunnel syndrome (CTS) [13,14]. The main risk determinants for this syndrome are rapid work pace with repetitive motion patterns, and insufficient recovery time associated with mechanical pressure concentrations. The main limit of existing studies is that most of them are conducted only through anamnesis collection and questionnaires, but only few integrate this information with physical examination and instrumental investigation [4,10].

In CTS, the compression of the surrounding soft tissues structures on the median nerve is responsible for its shape alteration. This effect results in a reduction of nerve volume at the site of compression and increased size proximal (and sometimes distal) to the compression. Thus, the increased cross-sectional area of the median nerve is the most commonly CTS ultrasound criterion for diagnosis [15].

Ultrasound imaging (UI) detects wrist changes in an early and often asymptomatic phase, but it has never been used in field studies on loggers [15,16,17]. To detect the presence of tenosynovitis, sonography relies on an assessment of retinaculum hypoechoic thickening and decreased tendon motion under dynamic maneuvers. Color and power Doppler can also be of use in the detection of neoangiogenesis in pathologic synovia proliferation. Tendinitis diagnosis, instead, mostly focuses on tendon enlargement and a hypoechoic or anechoic appearance with loss of the fibrillary pattern of the tendon, with the possible presence of calcifications [18]. The diagnosis of CTS with ultrasound is based on the measurement of the cross-sectional area of the median nerve proximal and distal to the carpal tunnel, as is has been observed to be higher in case of median nerve injury [19].

The main objective of our study has been evaluating if the slight morphological changes observed in a sample of chainsaw operators in 2017 showed an evolution to overt diseases after a two-year period in order to provide a clinical interpretation to the slight changes observed (adaptive vs. pathologic). Other objectives have been studying some risk determinants for these diseases such as age, working experience, and performing a secondary job.

## 2. Materials and Methods

This follow-up study involved 38 male forestry workers employed in tree felling, randomly selected among those employed in the Sicilian Forestry Department (state forestry corps), and located in Enna. The workers were contacted by email by the health and safety manager of the company. Among the 65 loggers invited to participate, 40 accepted and 21 declined. The response rate was 77.35%. Selection criteria were: a minimum of 3 years of experience as chainsaw operator; no musculoskeletal surgery or diagnosis of upper limb disease in the last 3 years; and to be aged between 18 and 70. The study population was selected in 2017 and followed up to 2019. Before participating, workers were informed about the approach and the aims of the study and provided informed consent to participation. The investigation was carried out at the workplace and lasted 5 days in November 2017 and 4 days in July 2019. According to EU criteria, workers were defined “old” if aged ≥55 and “young” if aged <55 years [20].

The study was approved by the Ethical Committee of the University Hospital G. Martino of Messina and respected the declaration of Helsinki.

The study protocol included:Personal data collection: age, Body Mass Index (BMI), dominant hand, working experience (years), sports/hobbies, type of contract; existence of a second job. In particular, for the contract type we considered the number of working days per year, while the second job, if present, was labeled as primary sector (extraction and production of raw materials), secondary sector (manufacturing), tertiary sector (services).Anamnesis has been collected through a questionnaire addressed at upper limbs symptoms with a hand chart (Figure 2). Hand chart included the radio-ulnar distal joint, and each participant was asked to report the presence or absence of four different symptoms: numbness/tingling, burn, ache, weakness [21,22]. Participants were also required to quantify the severity of the symptoms on a 0–10 scale, to specify their duration and clarify whether the symptoms appeared during working activity. Based on these data, we identified the subject showing symptoms related to the median nerve impairment in case of responses in the first three fingers, in the last two weeks before the data collection.Physical examination of the upper limbs, including Tinel’s and Phalen’s maneuvers performed on both wrists in order to detect carpal tunnel syndrome [23]; results were encoded in a dichotomous way as negative or positive regarding the presence of wrist disorders.UI of the hand-wrist area, performed with portable ultrasound device (Venue Scan, Venue 40, GE Healthcare^®^, Chicago, IL, USA) that allowed the researchers to store the examinations for further evaluation and discussion with other hospital experts in the final evaluation of all cases. A trained sonographer performed the wrist ultrasound. According to the European Society of Musculoskeletal Radiology’s protocol (ESSR) [23], acoustic windows were investigated in both the dorsal wrist, focusing on the extensor tendons and the radiocarpal joint, and the ventral wrist, focusing on the flexor tendons and the median nerve (Table 1).

In order to describe the severity of changes detected with the UI in tendons, we analyzed each tendon using the following classification [25]: tendinitis: (0), absence of tendinitis, no pathological changes; (1) presence of hypoechoic areas in the tendon’s structure or increase in the tendon’s size; (2) presence of both an increase in size and of hypoechoic areas; (3) presence of calcific metaplasia; tenosynovitis: (0) absence of tenosynovitis, no pathological changes; (1) minimal increase in the anechoic signal of the synovial space; (2) medium increase in the anechoic signal of the synovial space; (3) large increase in the anechoic signal of the synovial space; regarding the median nerve, the criteria for diagnosis of CTS was ultrasound evidence of enlargement of the diameter of median nerve in proximal carpal tunnel, with ≥0, 10 cm^2^ as a threshold. We measured the cross sectional area at the carpal tunnel’s inlet and we set the pathology threshold at ≥0, 10 cm^2^, according to the literature [25,26,27,28].

The same examinations have been conducted in both years 2017 and 2019.

For the statistical elaboration of the data, we used package SPSS PC version 23.

We used the Chi-Squared and Wilcoxon’s test to compare 2017’s questionnaire, hand chart, and US examination results to 2019’s. We compared the prevalence of wrist disorders considering the type of contract, the age, and the secondary job of the study’s subjects using the Chi-Squared test. Statistical significance was set at *p* ≤ 0.05 for all statistical tests.

## 3. Results

The present study involved a group of 38 male forestry workers, aged between 47 and 66 years, of which, both in 2017 and 2019, 14 (36%) were under 55 years, whilst 24 (63%) were over. Mean BMI was 27.8 in 2017 and 27.6 in 2019. Among the workers, 97.3% were right handed. The mean working experience was 27.52 years in 2017, and 29.39 in 2019. Only three workers reported having hobbies (trekking, dance) or playing sports (soccer, running). As for the working contract, three had a permanent job, nine had a temporary job of 151 days per year and 26 had a temporary job of 75 days per year (Table 2). Nothing has changed in 2019.

Among them, in 2017, 12 subjects worked only in forestry, while 14 declared a second job in the primary sector (mostly as farm workers), nine in the secondary sector (mostly as construction workers), and three in the third economic sector. In 2019, 4 subjects worked only in forestry, while 18 declared a second job in the primary sector, 13 in the secondary sector, and three in the tertiary sector. In 2019 we registered a statistically significative reduction of forestry workers without another job as it decreased from 12 (32.5%) in 2017 to 4 (10.5%) in 2019 (*p* = 0.005).

The differences between 2017 and 2019’s questionnaire results were statistically significant with a *p* = 0.020. In particular, according to the dominant hand’s questionnaire results, in 2017 the minority of the subjects, 10 (26.3%), declared the presence of symptoms related to possible wrist disorders with numbness/tingling and weakness being the most reported. In 2019, 19 workers (50%) reported presence of symptoms. Based on our classification criteria of the hand chart, in 2017, five (13.1%) workers resulted positive for wrist disorders, while in 2019 the number increased up 9 (23.6%), without statistical significance. The differences between 2017 and 2019 non-dominant hand’s questionnaire results were not statistically significant despite a slight increase in the report of symptoms from 8 (21%) subjects in 2017 to 11 (28.9%) subjects in 2019 (Figure 3). According to our classification criteria of the hand chart, for the non-dominant hand, the number of positive workers did not change between 2017 and 2019.

In both 2017 and 2019, the physical tests were positive in 10% of the subjects for both the dominant and the non-dominant hand. Subjects positive in 2019 were not the same subjects of 2017. UI revealed an overall increasing in the prevalence of tendinopathies (tendinitis and/or tenosynovitis) from 2017 to 2019. Considering all ultrasound compartments, the prevalence of tendinopathies in the dominant hand was 55.2% in 2017 and 63.1% in 2019 (*p* > 0.05), while for the non-dominant hand it was 47.3% in 2017 and 78.9% in 2019 (*p* = 0.04). The prevalence of tendinitis in the dominant hand was 36.8% in 2017 and 47.3% in 2019 (*p* > 0.05), while for the non-dominant hand it was 31.5% in 2017 and 60.5% in 2019 (*p* = 0.011). The prevalence of tenosynovitis in the dominant hand was 23.6% in 2017 and 47.3% in 2019 (*p* = 0.03), while for the non-dominant hand it was 23.6% in 2017 and 41.1% in 2019 (*p* > 0.05) (Table 3).

Comparing 2017 to 2019, the tendinitis of the dominant hand in 2019 resulted in nine new findings and nine already known cases as five were not pathological anymore, for 18 findings. In the non-dominant hand, we observed 14 new findings and 9 already known tendinitis as 3 were not pathological anymore. The tenosynovitis of the dominant hand in 2019 resulted in 10 new findings and 8 already known cases, as one was not pathological anymore. In the non-dominant hand, we observed seven new findings and nine already known tenosynovitis. Figure 4 shows a tendonitis in a first compartment, while Figure 5 shows a tenosynovitis in the fourth compartment. The prevalence of wrist tenosynovitis of the non-dominant hand resulted to be higher in younger workers.

At the UI, in 2017, the median nerve cross-sectional area at the carpal tunnel’s inlet reached pathological levels (≥0, 10 cm^2^) in 32% of the workers’ dominant hand and in 25% of the worker’s non-dominant hand. In 2019, we noted an increase of workers with enlarged cross-sectional area both in the dominant hand (81.5%) and in the non-dominant hand (55.2%). Of the 31 workers with an enlarged cross-sectional area at the dominant hands, 20 were new findings and 11 were already found in 2017. The differences between 2017 and 2019’s ultrasound results were statistically significant in both dominant hand with a *p* < 0.001 and non-dominant hand with a *p* = 0.008 (Table 4).

We observed a prevalence of cross-sectional area exceeding 10 square millimeters in 50% of the population aged ≥55 years, and only in 23% of the younger. However, this result did not reach statistical significance (*p* = 0.12). We observed a similar increase in median nerve pathology prevalence in the workers with greater working experience.

Performing a secondary job and working experience duration did not significantly affect the prevalence of wrist impairment (both neuropathies and tendinopathies) in these workers.

Moreover, we observed a higher prevalence in median nerve pathology of the non-dominant hand in workers with a 151 day contract, compared to the workers with a 75 day contract (*p* = 0.04). The prevalence was even higher in workers with a permanent job, even if this is not supported by statistical significance, possibly because of the small size of the group.

## 4. Discussion

The present study aimed to evaluate if the slight wrists’ morphological changes observed two years before in a sample of chainsaw operators were the first sign of a wrist disease or an adaptive condition not destined to evolve in an overt disease, considering also possible confounders such as age, working experience and performing a secondary job.

Our study results show an overall increase of tendinopathies and CTS in 2019 compared to 2017. This suggests that their wrist’s well-being should be given special consideration during occupational health medical examination as it can worsen or appear in a short amount of time.

In literature, there are few specific studies on upper limbs MSD among forestry workers. Besides, none of these combined anamnestic questionnaires, clinical evaluation, and UI detection, as proposed by our study design. In addition, most of the literature does not distinguish the main pathologies in tendinopathies and CTS, but only focuses on the pain of the upper limbs. Our results showed an increased prevalence of upper limbs disorders (21–50%), conversely to the range of 20 to 35% found in previous studies. This is probably due to our different approach considering not only pain but also other symptoms and morphological evidences [7,9,29].

UI revealed higher prevalence of tendinopathies if compared with the findings of Violante et al. [30] who studied the tendons of 160 meat production workers and found that 60 of 256 (about 24%) hands were affected. In the same way, this data suggest that chainsaw operators can have a higher risk of developing wrist impairment.

Interestingly subjects with CTS in 2017 were the same in 2019, and subjects with tendinopathies were different between 2017 and 2019. This is probably due to the intrinsic characteristics of these two different pathologies. In fact, CTS is considered a chronic syndrome, while tendinopathies often occur and resolve in short time. The recovery time for tendinitis is several days to six weeks, depending on whether treatment starts with early presentation or chronic presentation, while treatment for tendinosis recognized at an early stage can take from eight weeks to six months, depending on the characteristics [31].

Nevertheless, compared to our previous study on a population of 40 milking parlor workers [32], our UI results showed that the dominant hand’s median nerve cross cross-sectional area is more frequently affected in milkers (55%) than in forestry workers (32%) in 2017. In 2019, we found that the dominant hand’s median nerve cross-sectional area became more frequently affected in forestry workers (81.5%). Of the 31 subjects that ended up positive to the ultrasound in 2019, we found 20 new workers with pathological cross-sectional area, while all the 11 findings in 2017 remained positive. The questionnaire results are in line with UI results, as it was found positive in 25% of milkers and in 13.1% of forestry workers in 2017 and 23.6% in 2019.

The average age of our sample (50–55 years old) and consistent working experience probably influenced our results. Those workers who had longer experience in forestry are physically stronger and have a better understanding of the tasks’ procedures, despite their age and years of physical repetitive movements exposure, which are the main causes of occupational upper limb disorders [4,33]. The secondary job may also have influenced partially the prevalence of upper limbs MSD, in particular for those who experienced a secondary work in the tertiary sector.

According to the literature [34,35], our data showed a higher prevalence of median nerve impairment in the oldest population. Moreover, it does not surprise that this increase in prevalence is also present in those workers with greater working experience, as those are also the oldest workers.

It is interesting to underline that the population under 55 years seemed to be more at risk of developing tendonitis. This is probably due to the fact that, based on health surveillance outcomes, the older workers suffering different muscle skeletal diseases are assigned to less risky jobs, therefore with a lower risk of developing and occupational disease than their younger colleagues.

The use of portable ultrasound device is a novelty in the forestry sector and it was suitable and useful to study a consistent number of subjects at the workplace. The introduction of an imaging exam in a screening approach is innovative, as most studies have mostly focused only on anamnestic questionnaires and/or nerve conduction studies [14,21,35,36]. The absence of a power Doppler signal function in the portable US device should, however, be noted as a possible limitation of the study, as it could provide additional information on the presence of tenosynovitis.

In addition, although the usefulness of the portable ultrasound in detecting CTS signs, its sensibility is known in literature to be moderate (about 70%) and, to date, there is still not established/universal anatomical cut-off for the diagnosis [37,38]. Lastly, our research only focused on CTS and tendinopathies, but further studies focusing also on the Raynaud syndrome are desirable.

Moreover, involving seasonal workers may have introduced a confounding factor, due to a difficulty in evaluating the risk of developing wrists’ MSDs truly associated to the forestry tasks and the one related to the secondary job. In fact some of the workers who participated to the study had a secondary job in the primary or secondary sector, which are demonstrated to be highly associated to the risk of developing MSDs. Nevertheless we also noticed that in 2019 we registered a statistically significative reduction of forestry workers reporting a secondary job (from 12 in 2017 to 4 in 2019).

Lastly, the small sample size could have affected our results, in particular the not statically significant changes related to the cross-sectional area of the median nerve we noticed in the follow-up period.

## 5. Conclusions

The present study allows drawing interesting conclusions. Although not all the differences were statistically significant, we registered an overall increasing in wrist disorders in our population, without any particular influence by age, working experience and performing a secondary job.

Despite the specific preventive interventions adopted by the Sicilian Forestry Department, such as a proper posture training and the implementation of ergonomic designed chainsaws [2], the two-year follow-up survey brings to the conclusion that these measures were not fully sufficient or adequate or both.

Further studies could demonstrate if these preventive approach needs to be improved or applied for a longer time.

## Figures and Tables

**Figure 1 ijerph-18-07217-f001:**
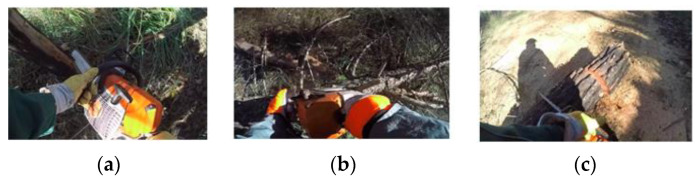
A worker performing the 3 main tasks of tree felling (**a**), tree delimbing (**b**), and tree bucking (**c**).

**Figure 2 ijerph-18-07217-f002:**
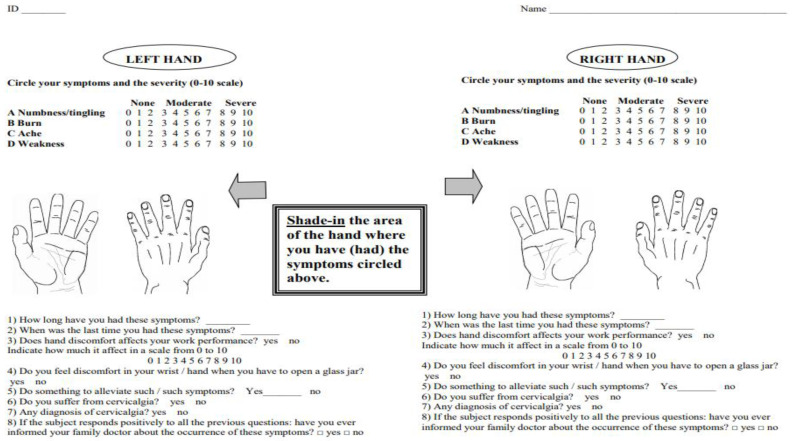
Hand Chart.

**Figure 3 ijerph-18-07217-f003:**
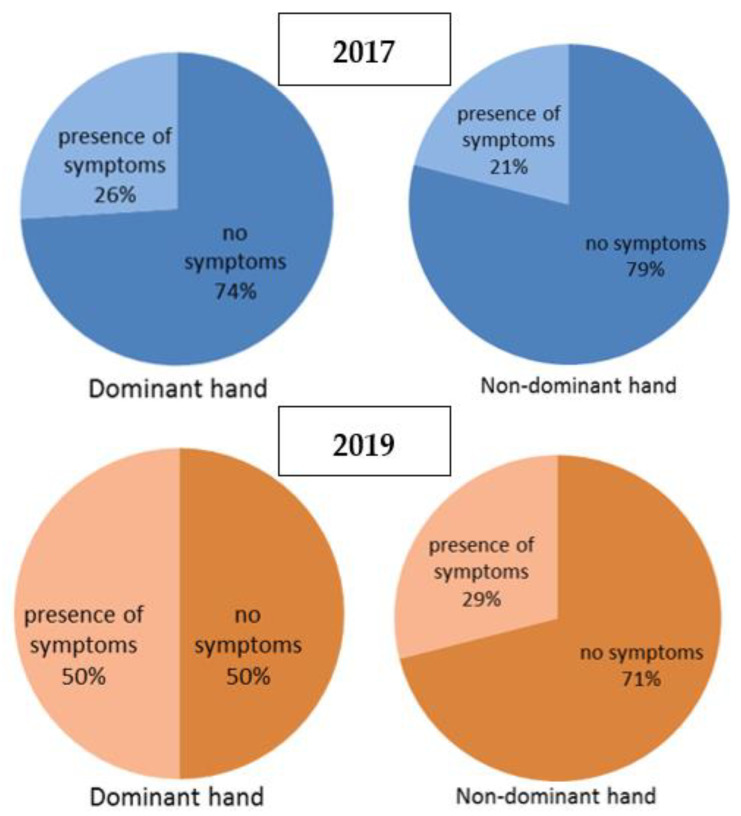
Symptom’s prevalence in dominant and non-dominant hand in 2017 and 2019.

**Figure 4 ijerph-18-07217-f004:**
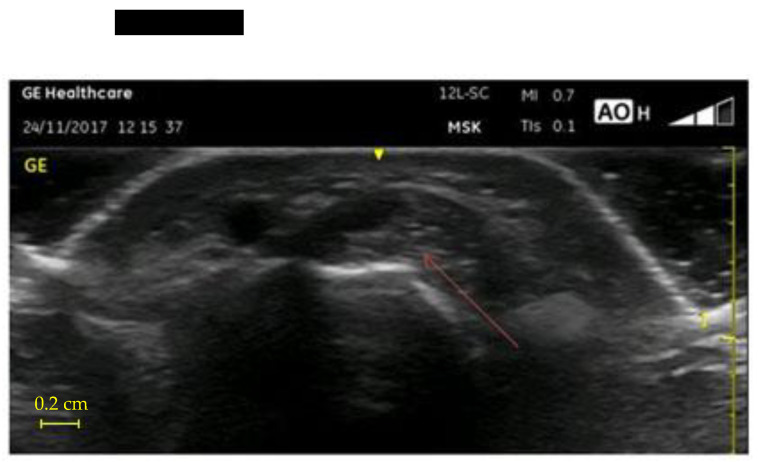
Ultrasonographic transverse image of the abductor pollicis longus and the extensor pollicis brevis, showing an increase in size and the presence of hypoechoic areas in the structure of the tendons.

**Figure 5 ijerph-18-07217-f005:**
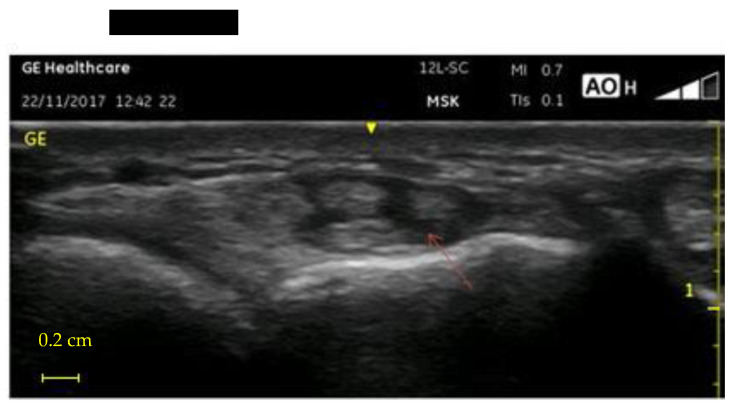
Ultrasonographic transverse image of the extensor digitorum communi tendons, showing a hypoechoic thickening of the synovial sheath.

**Table 1 ijerph-18-07217-t001:** Acoustic windows of the dorsal and ventral wrist [24].

US Dorsal Scans	Heading
Distal radioulnar joint	
I compartment	Retinaculum
	Abductor pollicis longus tendon
	Extensor pollicis brevis tendon
II compartment	Retinaculum
	Extensor carpi radialis longus tendon
	Extensor carpi radialis brevis tendon
III compartment	Retinaculum
	Extensor pollicis longus tendon
IV compartment	Retinaculum
	Extensor digitorum communis tendons
	Extensor indicis proprius tendon
V compartment	Retinaculum
	Extensor digiti quinti proprius tendon
VI compartment	Retinaculum
	Extensor carpi ulnaris tendon
Radiocarpal/midcarpal joint	
**US Ventral Scans**	
I compartment	Flexor carpi radialis tendon
II compartmentCarpal tunnel	Retinaculum
	Median nerve
	Flexor pollicis longus tendon
	Flexor digitorum superficialis tendons
III compartment	Guyon tunnel
IVcompartment	Flexor carpi ulnaristendon

**Table 2 ijerph-18-07217-t002:** Individual characteristics of study subjects.

Heading	All Subjects (*n* = 38), *n* (%) in 2017	All Subjects (*n* = 38), *n* (%) in 2019	*p* Values
Age (years), mean ± SD	52.7 ± 0.5	53.34 ± 0.5	*p* > 0.05
BMI, mean ± SD	27.8 ± 3.5	27.58 ± 3.6	*p* > 0.05
Right-handed	37 (97.3%)	37 (97.3%)	*p* > 0.05
Having hobbies or sports	3 (7.8%)	3 (7.8%)	*p* > 0.05
Years in current job, mean ± SD Other jobs Primary sector Secondary sector Tertiary sector None	27.52 ± 6.1 14 (37.5%) 9 (22.5%) 3 (7.5%) 12 (32.5%)	29.39 ± 6.1 18 (47.3%) 13 (34.2%) 3 (7.8%) 4 (10.5%)	*p* > 0.05 *p* > 0.05 *p* > 0.05 *p* > 0.05 *p* > 0.05 *p* = 0.005
Contract type			
75 days per year	26 (68.4%)	26 (68.4%)	*p* > 0.05
151 days per year	9 (23.6%)	9 (23.6%)	*p* > 0.05
Permanent job	3 (7.8%)	3 (7.8%)	*p* > 0.05

**Table 3 ijerph-18-07217-t003:** Prevalence of tendinopathies (tendonitis, tenosynovitis or both), isolated tendinitis, and isolated tenosynovitis in 2017 and 2019.

	2017	2019	*p* Value	2017	2019	*p* Value
	Dominant Hand (*n* = 38)	Dominant Hand (*n* = 38)		Non-Dominant HAND (*n* = 38)	Non-Dominant Hand (*n* = 38)	
**Tendinopathies**	21 (55.2%)	24 (63.1%)	*p* > 0.05	18 (47.3%)	30 (78.9%)	*p* = 0.04
**Tendinitis**	14 (36.8%)	18 (43.7%)	*p* > 0.05	12 (31.5%)	23 (60.5%)	*p* = 0.011
**Tenosynovitis**	9 (23.6%)	18 (47.3%)	*p* = 0.03	9 (23.6%)	16 (41.1%)	*p* > 0.05

**Table 4 ijerph-18-07217-t004:** Prevalence of positive questionnaire, positive hand chart and cross-sectional area >0.10 cm^2^ in 2017 and 2019.

	All Subjects (*n* = 38), *n* (%) in 2017	All Subjects (*n* = 38), *n* (%) in 2019	*p* Value
Positive questionnaire			
Dominant hand	10 (26.3%)	19 (50%)	*p* = 0.020
Non-dominant hand	8 (21%)	11 (28.9%)	*p* > 0.05
Positive hand chart			
Dominant hand	5 (13.1%)	9 (23.6%)	*p* > 0.05
Non-dominant hand	6 (15.8%)	6 (15.8%)	*p* > 0.05
US cross-sectional area >0.10 cm^2^			
Dominant hand	12 (32%)	31 (81.5%)	*p* < 0.001
Non-dominant hand	10 (25%)	21 (55.2%)	*p* = 0.008

## Data Availability

Data supporting the findings of this study are available from the corresponding author on request.

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
