# Peer review of "Hand-Wrist Disorders in Chainsaw Operators: A Follow-Up Study in a Group of Italian Loggers"

_ijerph, 2021, doi:10.3390/ijerph18147217_

Round 1

Reviewer 1 Report

Specific comments

1. Lines 19 and 79: Although the adopted statistical methodology in the manuscript is interesting, this approach is not able to “identify the main risk determinants”. In order to do this task, for a future paper, authors may consider a statistical predictive model (i.e. regression model). Hence, this statement might be replaced by something like “study/understand some risk factors”.

2. Figure 1 is not cited in the text.

3. Which sampling method was used to perform the analyses? Simple random sampling? Authors may specify this information alongside with the described criteria “minimum of 3 years of experience…”. Further, how was the sample size selected? Is it enough to make inferences?

4. Lines 90-91: If a worker is exactly 55 years old, how does he/she will be classified?

5. Authors may provide a translated version (in English) of Figure 2.

6. Some reference(s) may be provided for Lines 135-139. Authors should explain why do they use/select these tests in their analyses instead of several available in the literature.

7. Lines 141-143: please amend the text according to lines 90-91 (how to classify a 55-years old worker)

8. Line 144: If the same workers were analysed in years 2017 and 2019, how do the average working experience difference is greater than 2? I mean, considering exactly two years apart, if the average in 2017 was 27.2, then the average in 2019 should be 29.2. In lines 89-90, authors stated that this investigation was carried out in November 2017 and July 2019 (less than two years apart, and hence the average should be smaller than 2).

9. Lines 156, 175-180, 205: Please use p instead of capital P.

10. Lines 160-166: Authors may provide the calculated p-values for all statistical tests.

Author Response

REV 1

  1. Lines 19 and 79: Although the adopted statistical methodology in the manuscript is interesting, this approach is not able to “identifythe main risk determinants”. In order to do this task, for a future paper, authors may consider a statistical predictive model (i.e. regression model). Hence, this statement might be replaced by something like “study/understand some risk factors”.

R: We thank the reviewer for the remark. We have already conducted a cross sectional study on the same sample (currently submitted to another scientific journal) that aimed to identify the main risk determinants, therefore we agree it is more appropriate in this case to use the statement “study some risk factors” in the lines 19 and 79. Please see modified version of the manuscript.

  1. Figure 1 is not cited in the text.

R:Thank you for the remark. A sentence has been added in line 45. Please see modified version of the manuscript, where Figure 1 is cited.

  1. Which sampling method was used to perform the analyses? Simple random sampling? Authors may specify this information alongside with the described criteria “minimum of 3 years of experience…”. Further, how was the sample size selected? Is it enough to make inferences?

R: The workers were contacted by e-mail by the Health and Safety manager of the company. Among the 65 loggers invited to participate, 38 accepted and 27 declined. The response rate was 58.46%. Currently literature on matter is spare of big workers’ sample due to the difficulties in conducting the examination in the field, as we did. The above information have been included in the paragraph and a specific sentence has been added (see modified version) at linen 94-96.

  1. Lines 90-91: If a worker is exactly 55 years old, how does he/she will be classified?

R:Than you for the remark. If a worker is exactly 55 years old, he will be classified as >55 years old. We also modified the manuscript in the line 103, where “old” if aged≥ 55 replaced “old” if aged> 55.(See modified version)

  1. Authors may provide a translated version (in English) of Figure 2.

R: Thank you for your remark. Yes, we agree with the reviewer that is better to have a translated document, so we replaced the figure with the English version. See modified version of the manuscript.

  1. Some reference(s) may be provided for Lines 135-139. Authors should explain why they use/select these tests in their analyses instead of several available in the literature.

R: Thank you for your remark. We already conducted other studies on different populations, so the described protocol has already proved his efficacy. We also added a specific reference (the 23rd) we think very likely can reinforce our choice, following your appropriate suggestion. Please see modified version of the manuscript at lines 147.

  1. Lines 141-143: please amend the text according to lines 90-91 (how to classify a 55-years old worker)

R: Thank you for the remarks. Yes, we changed the sentences according to the previous reply. See previous response.

  1. Line 144: If the same workers were analysed in years 2017 and 2019, how do the average working experience difference is greater than 2? I mean, considering exactly two years apart, if the average in 2017 was 27.2, then the average in 2019 should be 29.2. In lines 89-90, authors stated that this investigation was carried out in November 2017 and July 2019 (less than two years apart, and hence the average should be smaller than 2).

R:Thank you for the remarks. Yes, we actually checked our data and we reported it wrongly in the table: in 2017 the average working experience was of 27.52 and in 2019 it was 29.39. Please see modified version of the table 2 and lines 164.

  1. Lines 156, 175-180, 205: Please use p instead of capital P.

R: Thank you for the remarks. Yes, we modified the text followed your suggestion.

  1. Lines 160-166: Authors may provide the calculated p-values for all statistical tests.

R: Thank you for the remarks. We added the information, following your precious suggestion. Please see modified version of the manuscript at table 1.

Reviewer 2 Report

I do not understand why you have chosen a longitudinal study concept, since a cross-sectional study design would have also answered the prevalence of wrist disorders. Furthermore, I feel that there is no clear explanation for the increase of tendinopathies or carpal tunnel syndrome? It might be, that forest workers were more sensible to notice minimal change of disorders due to study participation. Or have there been any changes regarding the equipment /saws / work conditions or was the increase of 2nd jobs relevant ?

l.72: Please name neovascularisation/power doppler signal as a further ultrasound sign of  tendinopathy. PLease name this limitation in the discussion. I assume the portable ultrasound could not generate

l.154: „three“ vs. „3“ in comaprison to l. 152.

  1. 150-l. 154: As far as I understand, more workers needed an additional job in 2019 which might have contributed to more work load and thus more MSDs. Please comment on this in your limitations.

Table 3: How do you define tendinitis vs. tendinopathy.

  1. 218: possibly because of the small size 218 of the group. --> no discussion in the result section.

  1. 292: wrist

Author Response

REV 2

I do not understand why you have chosen a longitudinal study concept, since a cross-sectional study design would have also answered the prevalence of wrist disorders. Furthermore, I feel that there is no clear explanation for the increase of tendinopathies or carpal tunnel syndrome? It might be, that forest workers were more sensible to notice minimal change of disorders due to study participation. Or have there been any changes regarding the equipment /saws / work conditions or was the increase of 2nd jobs relevant?

R:

1) The project was set up in two main strands: the first developed in another article which addresses the problem of the prevalence of wrist pathologists in the population under study and a second (the present) which aims to verify whether in a working population who continued to be exposed to repetitive movements and awkward posture we could notice a worsening of musculoskeletal disorders. In particular, this article aims to answer the question: can early signs of morphological changes have an impact on the evolution of the disease?

In the modified version of the present article we modified the first aim in order to better explain to the reader our final goal. See lines 18-19 and 84 and 249.

2) The data we collected in the present study through the questionnaire are supported by ultrasound examinations. In fact, when we affirm that there are worsening, we refer to objective data (instrumental measurements).

In the modified version of this article we have inserted a sentence explaining how the same exams were carried out in both years. See line 156.

l.72: Please name neovascularisation/power doppler signal as a further ultrasound sign of  tendinopathy. PLease name this limitation in the discussion. I assume the portable ultrasound could not generate.

R:Thank you for the remarks. As you assumed, our portable ultrasound device did not have the power doppler function. Please see the modified version of the manuscript at lines 77-78 and  308-311 where we specified this possible limitation.

l.154: „three“ vs. „3“ in comaprison to l. 152.

R:Thank you for the remarks. Yes, we followed your suggestion. Please see modified version of the manuscript.

l.150-l. 154: As far as I understand, more workers needed an additional job in 2019 which might have contributed to more work load and thus more MSDs. Please comment on this in your limitations.

 R: Thank you for the remarks. We first added more information about this aspect at lines 178-180, and following your suggestion we also added some remarks in the limitations at lines 314-318. Please see modified version of the manuscript.

Table 3: How do you define tendinitis vs. tendinopathy.

R: Thank you for the remarks. Tendonitis refers to an inflammation of the tendon, which can be seen at ultrasound as degenerative changes in the tendon’s structure. Tendinopathy is a general term which encompasses different types of tendon injuries. In our study, we consider “tendinopathy” the presence of tendonitis, tenosynovitis or both.

218: possibly because of the small size 218 of the group. --> no discussion in the result section.

R: Thank you for the remarks. We cited the numerosity of the sample as possible limitation of the study at line 325-327.

292: wrist

R: Thank you for the remarks. We have corrected the typos. Please see modified version of the manuscript.

Reviewer 3 Report

Comments to the Authors

GENERAL COMMENTS REGARDING PAPER

This study has the well done theoretical and practical approaches and it can be instructional for practitioners and researchers. However, there are some issues that have not been addressed in the manuscript. Despite such issues if these are addressed adequately it may be acceptable for publication following a second review.

MAJOR COMPULSORY REVISIONS

ABSTRACT

MAJOR 01: pg. 1. “Our research aimed to: a)” evaluate or compare?

MAJOR 02: pg. 1. The keywords should be different the words used in the title to increase the odd of manuscript impact.

INTRODUCTION

MAJOR 03: pg. 1 – First sentence. Please, do a double check in the journal rules.

MAJOR 04: pg. 2 – Figure 1. The figure should be referenced in the text.

MATERIALS AND METHODS

MAJOR 05: pg. 4 – Statistical Analysis. Add the effect sizes for comparisons.

RESULTS

MAJOR 06: pgs. 5 and 6. This figure should be improved to make clear the main message to the reader.

DISCUSSION

MAJOR 06: Insert some Practical Applications for the readers.

MINOR ESSENTIAL REVISIONS

MINOR 1: pg. 4. Typo: cm2 (special character)

Author Response

REV 3

GENERAL COMMENTS REGARDING PAPER

This study has the well done theoretical and practical approaches and it can be instructional for practitioners and researchers. However, there are some issues that have not been addressed in the manuscript. Despite such issues if these are addressed adequately it may be acceptable for publication following a second review.

MAJOR COMPULSORY REVISIONS

ABSTRACT

MAJOR 01: pg. 1. “Our research aimed to: a)” evaluate or compare?

 R:Thank you for the remarks. According to this remark in addition to the ones of the other reviewers we actually decided to reword the sentence for a better understanding of the readers. Please see modified version of the manuscript lines 18-19, 84 and 249.

MAJOR 02: pg. 1. The keywords should be different the words used in the title to increase the odd of manuscript impact.

R:Thank you for the remarks. Yes, we followed your suggestion and added three more key words:ultrasound investigation,CTS and hand chart. Please see modified version of the manuscript with changed key words.

INTRODUCTION

MAJOR 03: pg. 1 – First sentence. Please, do a double check in the journal rules.

R: The authors don’t understand the remark. We kindly ask the reviewer to better explain the comment.

MAJOR 04: pg. 2 – Figure 1. The figure should be referenced in the text.

R: Thank you for the remark. A sentence has been added in line 44 and the figure 1 was cited. Please see modified version of the manuscript.

MATERIALS AND METHODS

MAJOR 05: pg. 4 – Statistical Analysis. Add the effect sizes for comparisons.

R: Thank you for the remarks. We added the information, following your precious suggestion. Please see modified version of the manuscript at table 1.

RESULTS

MAJOR 06: pgs. 5 and 6. This figure should be improved to make clear the main message to the reader.

 R :Thank you for the remark. We modified the figures, please see modified version of the manuscript.

DISCUSSION

MAJOR 06: Insert some Practical Applications for the readers.

  R: Thank you for the remark. Please see modified version of the manuscript at lines 257-259

MINOR ESSENTIAL REVISIONS

MINOR 1: pg. 4. Typo: cm(special character)

R:Thank you for the remarks. We have corrected the typos. Please see modified version of the manuscript.

Round 2

Reviewer 2 Report

I have no further points which need to be addressed